

# Optimal input DNA thresholds for genome skimming in marine crustacean zooplankton

Junya Hirai

Atmosphere and Ocean Research Institute, University of Tokyo, Kahiwa, Chiba, Japan

## ABSTRACT

Crustacean zooplanktons are key secondary and tertiary producers in marine ecosystems, yet their genomic resources remain poorly understood. To advance biodiversity research on crustacean zooplankton, this study evaluated the effectiveness of genome skimming, a method that assembles genetic regions, including mitogenome, from shotgun genome sequencing data. Because the small amount of DNA available is a limitation in zooplankton genetics, different input DNA amounts (1 pg–10 ng) were prepared for library construction for genome skimming using two large species: *Euphausia pacifica* (Euphausiacea) and *Calanus glacialis* (Copepoda). Additionally, *de novo* assembly was used to obtain long contigs from short reads because reference-guided assembly can not be applied to all crustacean zooplankton. Evaluation of the raw sequence reads showed increased proportions of high-quality and distinct reads (low duplication levels) for large DNA inputs. By contrast, low sequence quality and high sequence duplication were observed for $\leq 10$ pg DNA samples, owing to increased DNA amplification cycles. Complete mitogenomes, including all 37 genes, were successfully retrieved for $\geq 10$ pg (*E. pacifica*) and $\geq 100$ pg (*C. glacialis*) of DNA. Despite the large estimated genome sizes of these zooplankton species, only $\geq 1$ and $\geq 3$ M reads were sufficient for mitogenome assembly for *E. pacifica* and *C. glacialis*, respectively. Nuclear ribosomal repeats and histone 3 were identified in the assembled contigs. As obtaining sufficient DNA amounts ($\geq 100$ pg) is feasible even from small crustacean zooplankton, genome skimming is a powerful approach for robust phylogenetics and population genetics in marine zooplankton.

## INTRODUCTION

Marine zooplankton are key secondary and tertiary producers in pelagic ecosystems. Crustacean zooplankton, including copepods and euphausiids, play a crucial role in the ocean carbon cycle due to their large biomass, production of fecal pellets and vertical migrations (*Steinberg & Landry, 2017*). Marine crustacean zooplankton also exhibit high species richness, and molecular approaches have revealed cryptic diversity that conventional morphological classifications cannot identify (*Bucklin et al., 2007*; *Blanco-Bercial et al., 2014*). In addition, molecular approaches are effective tools for revealing intraspecific and interspecific molecular phylogenetic relationships in crustacean zooplankton (*Lee, 2000*;

Corresponding author
Junya Hirai, hirai@aori.u-tokyo.ac.jp

*Goetze, 2003*; *Vereshchaka, Kulagin & Lunina, 2019*). These molecular-based zooplankton studies have mainly been conducted using mitochondrial DNA sequences obtained by Sanger sequencing, which require PCR amplification of specific genetic regions. However, the high diversity of crustacean zooplankton frequently causes primer mismatches during PCR, resulting in poor amplification of genetic markers of crustacean zooplankton, especially for mitochondrial markers with high mutation rates such as the mitochondrial cytochrome *c* oxidase subunit I (COI) gene (*Hirai, Shimode & Tsuda, 2013*; *Bucklin et al., 2021*). In addition, only a limited amount of total DNA is obtained from small zooplankton species with small body sizes, leading to difficulties in PCR amplification. A whole mitochondrial genome (mitogenome) sequence would provide a more robust phylogenetic analysis of zooplankton (*Machida et al., 2006*); however, only a short fragment of the target region can be obtained by single Sanger sequencing.

Considering the high species diversity of crustacean zooplankton, mitogenomes have been deposited in public databases at a slower pace than those of other marine taxa (*Bucklin et al., 2018*). Zooplankton have high mitogenome variability, and gene rearrangements are common, making it difficult to amplify mitogenomes using long PCR methods (*Machida et al., 2004*). To obtain mitogenome sequences and other useful genes, even in taxa with complicated genomic structures, the genome skimming approach is a promising method that does not incur high sequencing costs (*Straub et al., 2012*). In genome skimming, the DNA of the target species is extracted. Fragmented DNA is used for library construction with an amplification step free from species- or region-specific PCR primers for shotgun sequencing (Fig. 1). Massive sequence data are obtained using high-throughput sequencing (HTS), and target genetic regions including mitogenome is retrieved through informatics process. Because mitochondrial DNA is abundant in genomes, only small HTS sequence reads are necessary for mitogenome assembly. A genome skimming approach has been applied to marine crustacean zooplankton with medium and large body sizes, including *Eurytemora affinis* (*Choi et al., 2019*), *Calanus* spp. (*Choquet et al., 2017*; *Weydmann et al., 2017*; *Smolina et al., 2022*), and *Euphausia crystallorophias* (*Kim et al., 2024*), proving to be an effective method for obtaining mitogenomes. However, genome skimming remains largely unexplored for smaller crustacean zooplankton species, and few studies have evaluated the effects of initial DNA amount on genome skimming in these organisms. Additionally, the bait method, which uses reference data to assemble mitogenomes, cannot be applied to all species because reference sequence data (*e.g.*, COI) obtained by conventional Sanger sequencing are unavailable for many species of crustacean zooplankton (*Bucklin et al., 2021*).

The rapid development of HTS technology and lowering costs are expected to enhance genome skimming, enabling a more efficient and cost-effective method for sequencing and obtaining genomic resources, including mitochondrial and nuclear genes from various crustacean zooplankton species. However, some zooplankton species' low DNA amounts and complex genomic structures can limit genome skimming. To evaluate the effectiveness of genome skimming in crustacean zooplankton for future evolutional and ecological studies, this study focused on the major species of euphausiids and copepods: *Euphausia pacifica* and *Calanus glacialis*. The 16,898 bp mitogenome was reported for *E. pacifica* by
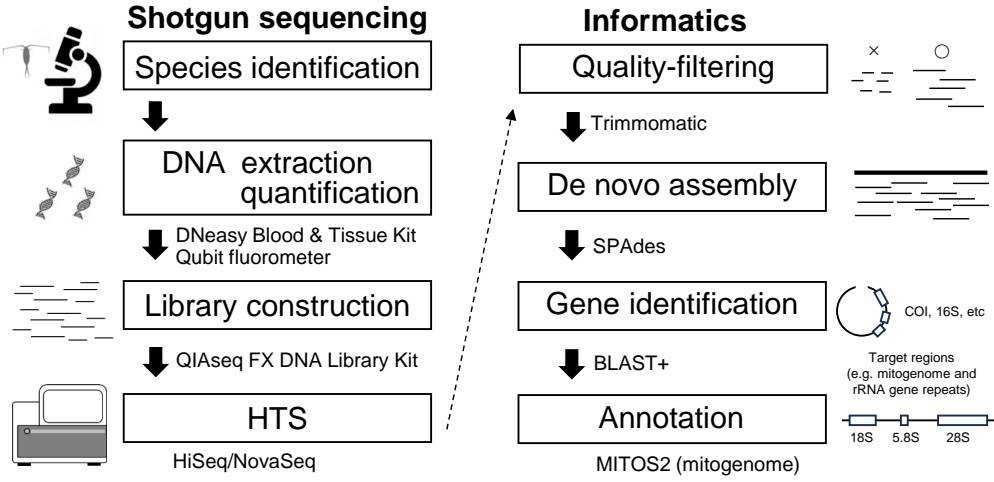

**Figure 1  Workflow for genome skimming in this study.**

*Shen et al. (2011)*. The 20,674 bp mitogenome sequence was first reported by *Choquet et al. (2017)*. However, *Weydmann et al. (2017)* obtained longer and more complete mitogenome sequences, split into two contigs (9,686 bp and 17,656 bp). For shotgun metagenomics for microbial communities, 1 pg DNA can be used for library construction, though data biases using 1 pg DNA are larger than those using ≥ 10 pg DNA (*Hirai et al., 2017*). Although the optimal amount of DNA for genome skimming is unclear for crustacean zooplankton, both *E. pacifica* and *C. glacialis* have large body sizes, which are useful for evaluating the effects of input DNA amounts on library construction. In addition, large genome sizes are estimated for the genus *Euphausia* (29.9–48.0 Gb; *Jeffery, 2012*; *Shao et al., 2023*) and *Calanus* (6.3–12.2 Gb) including *C. glacialis* of 11.8 Gb estimated genome (*McLaren, Sevigny & Corkett, 1988*). Large mitochondrial genomes were also reported especially for *Calanus* spp. with large non-coding and highly repeated regions (*Minxiao et al., 2011*; *Choquet et al., 2017*; *Weydmann et al., 2017*; *Kim et al., 2024*). These characteristics of complex genome structures are suitable for investigating the effectiveness of mitogenome assembly in crustacean zooplankton.

To evaluate optimal input DNA thresholds for genome skimming in marine crustacean zooplankton, this study used diluted DNA samples (1 pg–10 ng) for each species, and long contigs were assembled from shotgun sequencing data without mitochondrial reference sequences. Sequence quality was investigated for each sample, and the relationship among initial DNA amounts, sequencing depths, and genome skimming outcomes was assessed using the assembled mitogenomes. Additionally, the presence of nuclear genes, including ribosomal repeats, was investigated in the contigs.
**Table 1  Information about sampling events.**

| Species | Cruise | Station | Date | Lat. | Lon. | Depth |
|---|---|---|---|---|---|---|
| *Euphausia pacifica* | Hakuho-maru KH-14-3 | st. 12B | Jul. 27, 2014 | 53°70′N | 170°55′W | 0−549 m |
| *Calanus glacialis* | 2020 Garinko II cruise | st. A | Mar. 17, 2020 | 44°24′N | 143°25′E | 0−51 m |

## MATERIALS & METHODS

### Library construction and high-throughput sequencing

This study used single individuals of *E. pacifica* (Euphausiacea) and *C. glacialis* (Copepoda), collected from the Berring and Okhotsk Seas, respectively (Table 1). Bulk samples were preserved in 99% ethanol at −20 °C. Specimens of *E. pacifica* and *C. glacialis* were identified based on *Baker, Boden & Brinton (1990)* and *Chihara & Murano (1997)* and confirmed through mitochondrial gene sequences, including COI obtained in this study, distinguishing them from closely related species, including *C. marshallae*. The genome skimming workflow is depicted in Fig. 1. Species were identified using a stereomicroscope. Genomic DNA was extracted from an individual of each species using a DNeasy Blood & Tissue Kit (Qiagen) and eluted in 50 µL of Buffer EB (Qiagen). Due to a large body size, only part of the body was used for DNA extraction of *E. pacifica*. DNA concentration was measured using a Qubit fluorometer (Invitrogen). DIN (DNA integrity number) was measured using the Genomic DNA system on TapeStation 4200 (Agilent Technologies) to evaluate the extracted DNA's quality. Libraries for genome skimming were constructed using the QIAseq FX DNA Library Kit (Qiagen) for five different amounts of DNA for each species (10 and 1 ng and 100, 10, and 1 pg). These DNA amounts were selected to evaluate the effects of DNA inputs on genome skimming data for zooplankton with small DNA amounts. Although the QIAseq FX DNA Library Kit requires at least 20 pg DNA in the original protocol, a small amount of DNA up to 1 pg was used in this study to compare the different amounts of DNA inputs and to evaluate if the genome skimming method can be applied to small zooplankton or part of the body of specimens. Libraries were constructed basically according to the manufacturer's protocol. Three additional cycles were applied for the amplification step for 100 pg (17 cycles), 10 pg (20 cycles), and 1 pg (23 cycles) samples based on the results of preliminary experiments, though 10 ng (10 cycles) and 1 ng (12 cycles) samples were followed by the protocol. The details of the fragmentation time, adaptor dilution, and PCR cycles during the amplification step are provided for each DNA input in Table S1. The quality and concentration of the constructed libraries were measured using a D1000 system on TapeStation 4200 and Qubit. Equal library DNA amounts were pooled for HTS, except for the 1 and 10 pg samples of *C. glacialis*, for which sequence libraries were not successfully obtained. Raw 2 × 150-bp paired-end sequence reads were obtained using the Illumina HiSeq and NovaSeq systems.

## Mitochondrial genome

Adaptor sequences were trimmed and low-quality reads were removed using Trimmomatic *v*. 0.39 (*Bolger, Lohse & Usadel, 2014*) with default settings (ILLUMINACLIP:TruSeq3-PE-2. fa:2:30:10, LEADING:3, TRAILING:3, SLIDINGWINDOW:4:15, and MINLEN:36). Sequence quality and duplication levels were evaluated using FastQC (http://www.bioinformatics.babraham.ac.uk/projects/fastqc/). Quality-filtered samples were assembled using SPAdes (*Bankevich et al., 2012*). Contigs for mitogenomes were identified by a BLAST search against the reference mitogenome of *E. pacifica* (GenBank accession: EU587005; *Shen et al., 2011*) and *C. glacialis* (MG001883 and MG001884; (*Weydmann et al., 2017*) using BLAST+ (*Camacho et al., 2009*). The mitogenome of *C. glacialis* was registered in two separate contigs because of difficulties in assembly. This study investigated only contigs with ≥1,500 bp for mitogenome. The obtained contigs were annotated using MITOS version 2.1.9 (*Arab et al., 2017*; *Donath et al., 2019*). The results of *de novo* assembly using SPAdes were compared to those obtained by mapping method against the mitochondrial genome sequences of 1 ng samples in *E. pacifica* and *C. glacialis* using Geneious Prime 2019.0.4 (https://www.geneious.com) with the option of 'Medium-Low sensitivity' and up to five iterations. The lengths of the longest mitochondrial sequences were compared with different read numbers to determine the number of reads necessary for mitogenome assembly.

## Nuclear genes

Contigs containing nuclear rRNA genes (18S and 28S) were analyzed using BLAST+. In addition to the 18S and 28S sequences of *E. pacifica* and *C. glacialis*, internal transcribed spacer (ITS) sequences were added to the reference sequence sets. Histone 3 (H3), commonly used for phylogenetic analysis of crustacean zooplankton (*Cornils & Blanco-Bercial, 2013*; *Vereshchaka, Kulagin & Lunina, 2019*), was also investigated. Because there were no ITS sequences for *E. pacifica* and H3 for *C. glacialis*, the sequence data of species in the same genus were used as reference data for the BLAST search.

# RESULTS

## Sequence quality

The DNA concentrations were 15.0 ng/μL (*E. pacifica*) and 14.7 ng/μL (*C. glacialis*), and these extracted DNA showed DIN of 5.7 and 6.5 for *E. pacifica* and *C. glacialis*, respectively (Fig. S1). Libraries were successfully prepared for genome skimming of *E. pacifica*, and 2.77−9.98 M reads were obtained (Table S2). In *C. glacialis*, >6.96 M reads were obtained for samples with ≥100 pg DNA inputs. However, only 1.58 M and 419 reads were obtained for samples of 10 and 1 pg, respectively, owing to difficulties in preparing sequencing libraries for *C. glacialis* with low DNA amounts. The proportion of high-quality reads was 60.4−77.6% for *E. pacifica* and 25.3−79.9% for *C. glacialis* (Fig. 2A). These values tended to decrease from 100 to 1 pg, and the lowest quality reads were observed in the 1 pg samples for both species. The proportions of distinct reads, which are sequences after removing duplicate reads, were 87.9−88.2% for ≥100 pg DNA samples in *E. pacifica* (Fig. 2B) and decreased to 71.5% (10 pg) and 23.0% (1 pg). In *C. glacialis*, the proportion of distinct
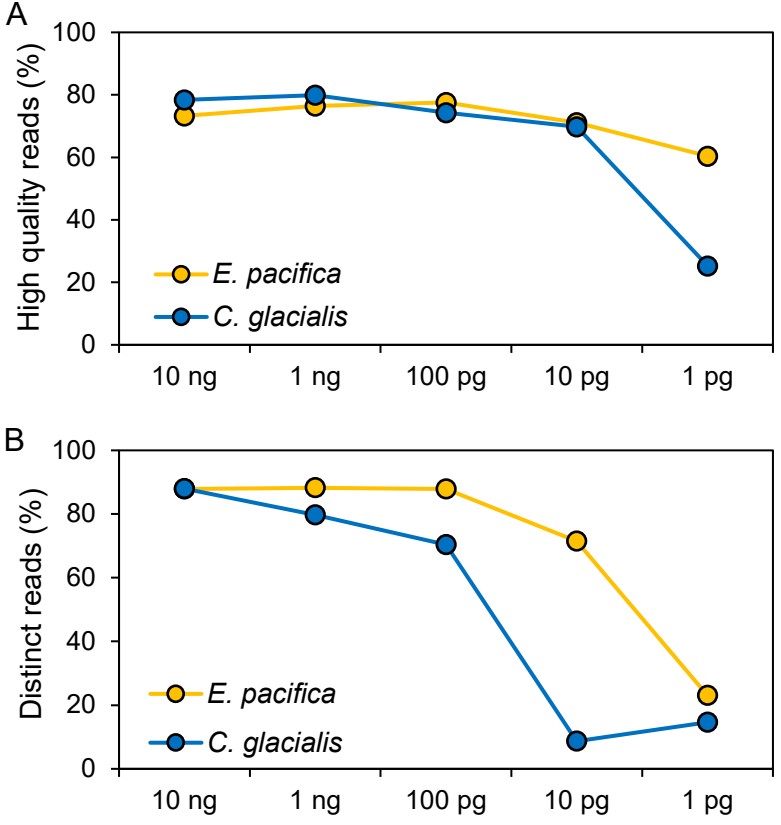

**Figure 2** **Quality of sequence reads.** (A) Proportions of high-quality sequence reads after the quality-filtering step. (B) Proportions of sequence reads that remained if duplicated reads are removed (distinct reads). The numbers of reads used for analyses are listed in Table S2.

reads decreased from 10 ng (88.0%) to 100 pg (70.4%), and only a small proportion of distinct reads remained in the 10 pg (8.7%) and 1 pg (14.6%) samples.

## Mitogenome sequences

In *E. pacifica*, almost full-length mitogenomes were obtained as a single contig for samples using 10 ng, 1 ng, 100 pg, and 10 pg of DNA (Fig. 3A). These sequences were identical to the mitogenome sequences mapped to the reference sequence, with high sequence coverage. All 37 genes were successfully annotated in the mitogenome contigs of *E. pacifica*. The terminal ends of the contigs were observed in the control region of *E. pacifica*, and overlapping regions were not detected at either end. In the sample using 1 pg of DNA, the longest contig identified as a mitochondrial sequence was only 1,869 bp, and the full genome sequence was not obtained.

Because the reference mitochondrial genome was separated into two parts in *C. glacialis*, two contigs were obtained for the mitochondrial sequences of *C. glacialis* in samples containing 10 ng and 1 ng of input DNA (Fig. 3B). As observed in *E. pacifica*, overlapping regions of the mitogenome with sufficient sequence coverage were identical between the *de novo* assembly and mapping methods. The annotation detected 37 genes in the contigs.

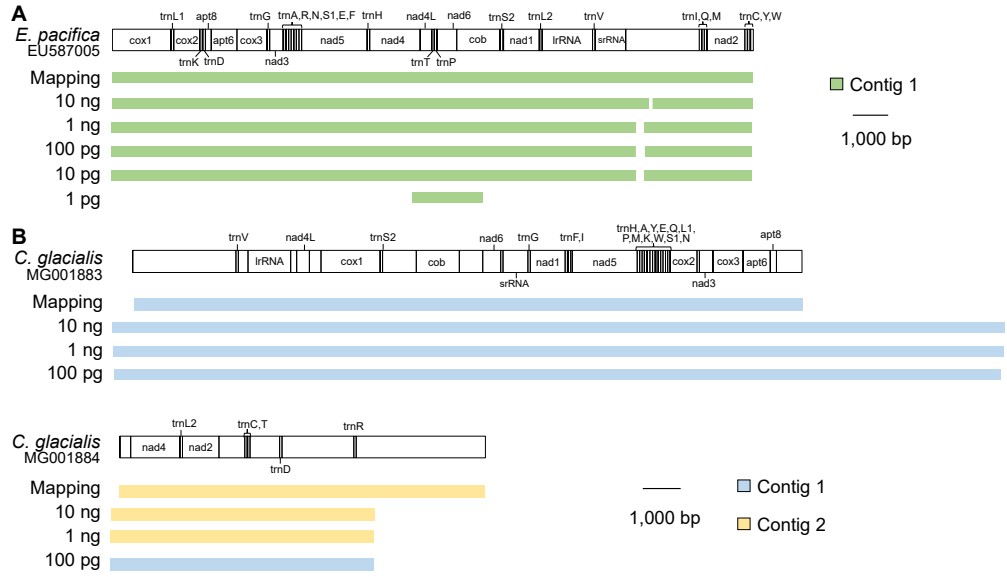

**Figure 3** **Mitochondrial genome sequences for different DNA inputs.** (A) *Euphausia pacifica*. (B) *Calanus glacialis*. Note that no >1,500 bp mitochondrial contigs were obtained for 10 and 1 pg samples of *C. glacialis*. Note that start and end points are different between reference sequences and assembled contigs.

However, the two copies of trnL2 reported by *Weydmann et al. (2017)* were not detected in this study's mitochondrial contigs of *C. glacialis*. In the sample of 100 pg with large sequence reads, two contigs were merged into a long contig of >30,000 bp, although the accuracy of the merged sites of the two contigs could not be evaluated. The terminal ends of the contigs were observed in the control regions, which were difficult to merge to recover a circular mitogenome. Sequence libraries were not successfully constructed for the 10 and 1 pg DNA samples from *C. glacialis*; no mitochondrial contigs >1,500 bp were obtained.

## Sequencing depth

To evaluate the sufficient number of sequence reads for mitogenome assembly, 100 pg samples with the maximum number of sequence reads were used for each species. In *E. pacifica*, almost the same length of the longest mitochondrial sequences were observed for $\geq 1$ M reads (16,642−16,709 bp; Fig. 4), and these sequences were identical. The longest mitochondrial contig was below 8,000 bp in the case of 0.5 M reads for assembly. Larger sequence reads were necessary to assemble mitochondrial sequences in *C. glacialis*, which has a longer mitochondrial genome than *E. pacifica*. The longest mitochondrial contig increased to 23,469 bp from 1 M to 3 M reads. Mitochondrial sequences of almost the same length were observed in 3−6 M reads. The separated mitochondrial contigs (Fig. 3) were merged, and the longest contig of 30,310 bp was retrieved using the maximum number of 6.38 M reads for assembly.

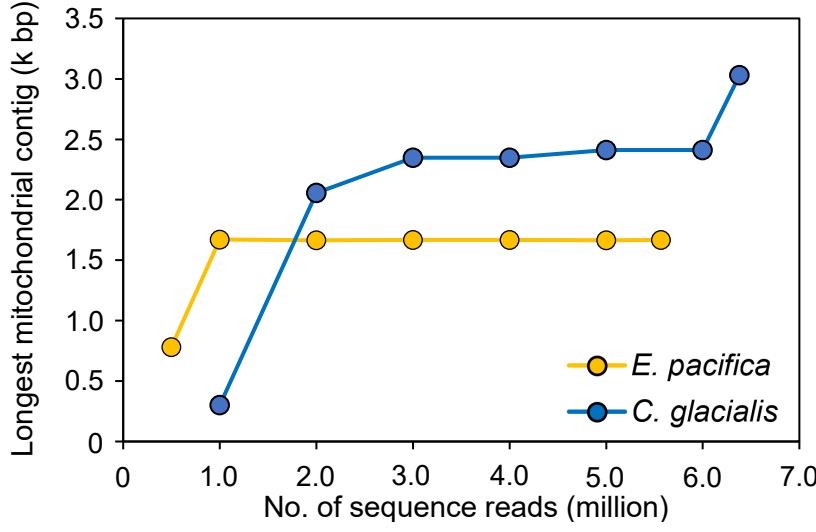

**Figure 4** **The longest mitochondrial contig in different numbers of sequence reads for assembly.** The samples with 100 pg input DNA are used for *Euphausia pacifica* and *Calanus glacialis*.

**Table 2** **The longest contigs for nuclear rRNA genes of 18S and 28S in each sample are listed.**

|  |  | 18S | 28S |
|---|---|---|---|
| *Euphausia pacifica* | 10 ng | 3,679 bp[*] | 4,426 bp |
|  | 1 ng | 5,369 bp[*] | 3,967 bp |
|  | 100 pg | 1,400 bp | 1,346 bp |
|  | 10 pg | 1,400 bp | 1,346 bp |
|  | 1 pg | 1,028 bp | 1,090 bp |
| *Calanus glacialis* | 10 ng | 1,289 bp | 1,927 bp |
|  | 1 ng | 2,075 bp[*] | 3,266 bp[*] |
|  | 100 pg | 2,457 bp[*] | 3,921 bp[*] |
|  | 10 pg | 1,930 bp[*] | 1,060 bp |
|  | 1 pg | no data | |

**Notes.**
[*]Contigs including internal transcribed spacer (ITS) regions.

## Nuclear genes

All samples, except for 1 pg of *C. glacialis*, contained nuclear 18S and 28S contigs (Table 2). Contigs including 18S and 28S were separated from all samples. Both 10 and 1 ng samples showed relatively long contigs for 18S and 28S in *E. pacifica*. The ITS sequences were included in the 18S rRNA contigs of these samples. In *C. glacialis*, relatively large contigs, including 18S and 28S, were detected in the 1 ng and 100 pg samples. In these samples, both the 18S and 28S contigs contained ITS regions. The 10 pg sample of *C. glacialis* had contigs of 18S and 28S, and ITS regions were included in the 18S contig. H3 was detected in all samples of *E. pacifica*; however, only samples of 10, 1 ng, and 100 pg contained H3 contigs in *C. glacialis*.
## DISCUSSION

Genomic resources, including mitogenomes and useful nuclear genes obtained by genome skimming, are useful for evolutionary and ecological studies. However, this approach has not been widely used in marine crustacean zooplankton. In this study, two major crustacean zooplankton species, *E. pacifica* and *C. glacialis* were selected to evaluate the effects of input DNA amounts on genome skimming, using diluted DNA from 1 pg to 10 ng. One of the goals of this study was to construct a simple workflow for genome skimming methods in marine zooplankton, including small crustaceans, without relying on reference sequence data. In the shotgun genome sequencing library construction, physical fragmentation and linker ligation methods tend to perform better than enzymatic approaches (*Knierim et al., 2011*; *Hirai et al., 2017*). Physical fragmentation of DNA is commonly used in zooplankton genome skimming (*Smolina et al., 2022*); however, not all laboratories have access to equipment for physical fragmentation, such as a Covaris focused-ultrasonicator. Enzymatic DNA fragmentation was used for library construction using a commercially available kit in this study. In addition, this study used the *de novo* assembly method (SPAdes) to construct contigs, although there are other useful methods, such as the bait method (*e.g.*, NOVOplasty; *Dierckxsens, Mardulyn & Smits, 2017*) and detections of target genes based on amino acid sequences (*e.g.*, MitoGeneExtractor; *Brasseur et al., 2023*). This study selected *de novo* assembly to target various regions including mitogenome and nuclear rRNA genes, because reference sequence data are not always available for various crustacean zooplankton species. The mitogenome sequences of *E. pacifica* and *C. glacialis* in this study were identical to those obtained by mapping to reference mitogenome sequences using different methods (*Shen et al., 2011*; *Weydmann et al., 2017*). As molecular-based methods are still being developed in marine zooplankton (*Bucklin et al., 2018*), the workflow in this study is effective for accumulating future mitogenome data for marine zooplankton. However, complete mitogenome data were not obtained in this study for *E. pacifica* or *C. glacialis* because control regions with repeat sequences (*Shen et al., 2011*; *Weydmann et al., 2017*) were difficult to assemble. Two contigs of *C. glacialis* reported by *Weydmann et al. (2017)* were merged into one contig in the 100 pg sample; however, the accuracy of the sequences in this merged region should also be confirmed. Although genome skimming retrieved all mitochondrial genes of *E. pacifica* and *C. glacialis*, combining this method with conventional Sanger sequencing is expected to yield complete mitogenome sequences, including control regions. Although large non-coding and highly repetitive regions are difficult to assemble based on shotgun sequencing data, complex regions can also be retrieved if long-read sequencers are used. Technical replicates were not analyzed for each DNA input in this study. However, identical sequences were observed among contigs of different DNA inputs, suggesting that the results of genome skimming are consistent.

This study implied that at least 100 pg DNA should be used for high-quality genome skimming data in crustacean zooplankton. Although some crustacean zooplankton are small, the total extracted DNA ranged from 600 pg to $2.4 \times 10^2$ ng in 90 species of small/large copepods in the Kuroshio Current regions off the coast of Japan. The DNA amount per nucleus ranges from 4.32 to 24.92 pg for marine calanoids, although smaller

values of 0.28–1.8 pg DNA per nucleus have been reported for Cyclopoida (*Wyngaard & Rasch, 2000*). A sufficient amount of DNA was extracted from adult females of small copepods in the genus *Paracalanus* (4.9–7.3 ng/μL) and *Spinocalanus* (2.3–8.3 ng/μL) (*Cornils, 2015*). Thus, the extraction method to obtain sufficient amounts of DNA is not a severe limitation for genome skimming in crustacean zooplankton. The necessary DNA input for shotgun sequencing depends on a library construction kit. For example, the NEBNext Ultra DNA Library Prep Kit (New England Biolabs) required five ng DNA (*Liu et al., 2012*) but was updated for as low as 100 pg DNA in the version II kit. The present study also showed that genome skimming data could be obtained by adding the amplification step for library construction using 1 and 10 pg of DNA inputs, below the minimum DNA amount (20 pg) in the original protocol using the QIAseq FX DNA Library Kit. The same result was also obtained for shotgun metagenome sequencing of microbial communities using different library construction kits of the KAPA Hyper Prep Kit and Nextera XT DNA Library Preparation Kit, and 10 pg was reported as a limit for low-biases library construction (*Hirai et al., 2017*). Sequence quality and diversity of sequences decreased in genome skimming data using a small amount of DNA input because of the high duplication levels induced by increased PCR cycles to obtain sufficient library concentration. However, sequence data obtained from a small amount of DNA can also be used for ecological studies of zooplankton in case of difficulty in accessing high concentrations of DNA for target species.

In a previous study of genome skimming, deep sequence data of approximately 178 and 175 M paired-end reads of 150 bp were used for the mitogenome and nuclear ribosomal repeats of *C. finmarchicus* and *C. glacialis*, respectively (*Weydmann et al., 2017*). Other genome skimming studies in marine crustacean zooplankton also used a massive amount of raw sequence reads: 52 M for *C. simillimus* (*Smolina et al., 2022*), 37.2 M for *E. crystallorophias* (*Kim et al., 2024*), and 46.9 M for *Tigriopus kingsejongenesis* (*Hwang et al., 2019*). However, deep-sequencing efforts are not always necessary for genome skimming, as mitogenomes and nuclear ribosomal repeats have been successfully assembled using 2–3 M reads for marine fish (*Hoban et al., 2022*). The comparison of different sequence reads for assembly in this study indicated that ≥1 and ≥3 M reads were enough to obtain mitogenomes for *E. pacifica* and *C. glacialis*, respectively. The minimum number of sequence reads required for mitogenome assembly may be associated with different sizes. The mitogenome size of *C. glacialis* is >27,342 bp (*Weydmann et al., 2017*), which is longer than that of other crustaceans, including *E. pacifica*, which has a 16,898 bp mitogenome (*Shen et al., 2011*). Although the genome size of *E. pacifica* has not been estimated, a large estimated genome size of >10 Gb was reported for species in Euphausiacea (*Jeffery, 2012*), and an 11.8 Gb genome size was estimated for *C. glacialis* (*McLaren, Sevigny & Corkett, 1988*). These genome sizes were larger than those of other zooplankton (*Gregory, 2024*), except for some crustacean species with uniquely large genome sizes, such as the 63.2 Gb genome of the Amphipoda *Ampelisca macrocephala* (*Rees et al., 2007*). The variation in copepod genome size is large, especially in Copepoda (*Wyngaard & Rasch, 2000*), and the minimum sequencing depth for genome skimming among species may differ. The amount of sequence data for assembly depends on the target genetic region and sequencing

methods such as sequence length. However, the results obtained using species with large genome sizes in this study suggest that several million sequence data of shotgun sequencing can be assembled for almost full mitogenome sequences and other nuclear genes, including ribosomal repeats.

In addition to mitogenome sequences obtained by genome skimming, sequences obtained from transcriptome data are useful for robust phylogenetic analysis of crustacean zooplankton, including euphausiids and copepods (*Lizano et al., 2022*; *Choquet et al., 2023*; *Iwanicki et al., 2024*). Whole-genome sequencing data have also been obtained for a limited number of crustacean zooplankton, including copepods and euphausiids, using long-read sequencers (*Choi et al., 2021*; *Shao et al., 2023*; *Du et al., 2024*; *Unneberg et al., 2024*). However, transcriptome and whole-genome analyses require high-quality DNA or RNA, and the costs of these analyses are higher than those of genome skimming. In this study, samples were collected almost 10 years ago (Table 1), and degradations of DNA were observed to some extent based on the results of DIN (*Kong et al., 2014*). Although the DNA quality of ethanol-preserved samples decreases over time (*Goetze & Jungbluth, 2013*), this study showed that DNA amount is an important factor for low-bias library construction of shotgun sequencing in case of moderate DNA quality. As the genome skimming approach can be applied to old zooplankton samples, it is also useful for specimens previously used for DNA barcoding (*Bucklin et al., 2021*). According to *Hoban et al. (2022)*, the total cost of genome skimming, including library construction, quantitation, and sequencing, is only approximately $31 per sample when multiplexed and sequenced using the Illumina NovaSeq system. Because sequencing technology is developing rapidly, genome skimming costs are expected to decrease further. Genome skimming data have advantages over conventional DNA barcoding with limited sequence length and primer mismatches. Thus, this method is expected to be widely used for future ecological and evolutionary studies of marine zooplankton.

## CONCLUSIONS

This study demonstrated that genome skimming can be a powerful approach for robust phylogenetics and population genetics in crustacean zooplankton. In genome skimming, it addressed the challenge of low total genomic DNA quantities in marine zooplankton using diluted DNA from the large zooplankton *E. pacifica* and *C. glacialis*. Although previous studies mainly focused on medium and large crustacean zooplankton, this study showed that at least 100 pg of DNA is required for genome skimming, and 1 ng is the optimal amount of DNA for obtaining robust data on mitogenome and nuclear genes. Understanding the interplay of DNA inputs and genome skimming techniques is expected to be crucial in future conservation and biodiversity studies for diverse organisms, including marine zooplankton.

## ACKNOWLEDGEMENTS

I thank all the members of the cruises on RV Hakuho-maru and Garinko II for their assistance with sample collection. I also thank M. Noda for assistance with laboratory works.

### Funding

This work was supported by a research project grant from the grants provided by the Environmental Research and Technology Development Fund (JPMEERF20224R03) of the Environmental Restoration and Conservation Agency provided by the Ministry of the Environment of Japan. The funders had no role in study design, data collection and analysis, decision to publish, or preparation of the manuscript.

### Grant Disclosures

The following grant information was disclosed by the author:
The Environmental Research and Technology Development Fund of the Environmental Restoration and Conservation Agency provided by the Ministry of the Environment of Japan: JPMEERF20224R03.

### Competing Interests

The author declares there are no competing interests.

### Author Contributions

- Junya Hirai conceived and designed the experiments, performed the experiments, analyzed the data, prepared figures and/or tables, authored or reviewed drafts of the article, and approved the final draft.

### DNA Deposition

The following information was supplied regarding the deposition of DNA sequences:
The raw sequence data for genome skimming are available at NCBI/EBI/DDBJ Sequence Read Archive: PRJDB19046.

### Data Availability

The data on the main results, including all assembled contigs, contigs for the mitogenome obtained by de novo assembly and mapping methods, contigs for nuclear rRNA repeats, and results of annotations, are available at Figshare: Hirai, Junya (2024). Effects of input DNA amounts on genome skimming in marine crustacean zooplankton. figshare. Collection. https://doi.org/10.6084/m9.figshare.c.7502001.v1.

### Supplemental Information

Supplemental information for this article can be found online at http://dx.doi.org/10.7717/peerj.19054#supplemental-information.

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
