# Peer review of "Optimal input DNA thresholds for genome skimming in marine crustacean zooplankton"

_PeerJ, doi:10.7717/peerj.19054_

## Round 0.1 · original submission · Major Revisions

Dear author,
I have now received three reviews of your manuscript, two of which recommend a major revision and one of which favors a minor revision. All three reviewers have raised multiple questions and comments, and I believe that addressing them can really improve the usefulness of the manuscript. I therefore invite you to revise your manuscript in line with the reviews you have received. Please take particular care to address concerns such as the lack of methodological details that allow replication of your procedure, the provision of raw data (as raised by reviewers 1 and 3), and the inclusion of gel images of DNA (reviewer 1). As this is a paper that has the potential to set methodological standards in the field of zooplankton, it is important that all issues are clarified to help the community avoid future pitfalls. Also, please ensure that a brief description of species identification (with references) is included in Materials and Methods, as highlighted by reviewer 1, as these details are always of interest to “zooplankton people”, myself included. At this point I see no need for linguistic corrections, but there is the same typo in the supplementary tables (1 and 2), so please change Euphasia to Euphausia. Also, the frequently used “de novo” should be italicized throughout the manuscript. Otherwise, the language is impeccable.

Please include with the revised version of your manuscript a detailed response to the comments received (point by point) with clear answers and references to corrections.

Thank you for submitting your work to PeerJ, with kind regards,

Olja Vidjak

·

Basic reporting

1.1. The paper has clear, unambiguous, professional English language through out.

A. Please clarify the term “genome skimming” further, as it is used throughout the paper but only briefly defined.
B. Additionally, the term “genomics” needs clarification to avoid confusion with “mitogenome.” Genomics refers to both the mitogenome and nuclear parts of the genome.

1.2. Intro & background to show context.

Just minor corrections / suggestions

Line 52: from zooplankton species with small body size
Line 75: Add with lowering cost, The rapid development of HTS technology and lowering cost…
1.3. Literature well referenced & relevant

Yes, all the paper is well referenced, except I think there was a correction in Choquet et al. (2017), where they reported a full mitogenome of C. glacialis. However, it turns out it wasn’t complete, as indicated in Weydmann et al. (2017). (Please double-check.)

1.4. Structure conforms to PeerJ standards, discipline norm, or improved for clarity.
Yes
1.5. Figures are relevant, high quality, well labelled & described.

Excellent figures and very easy to understand.

Comment: Please include the gel images or tape station images of DNA at different concentrations prior to library preparation as supplementary material or in the paper. This could give readers an idea of the quality of the DNA extracted from the samples.

1.6. Raw data supplied (see PeerJ policy).

Raw data was not provided nor uploaded in NCBI Database. I advise that the authors upload the data used in NCBI database.

Experimental design

Yes, the paper is original primary research within the aims and scope of the journal. This paper also had the research question well defined, relevant, and meaningful.

There are some comments regarding the experimental design. While the overall experiment is interesting and makes a valuable contribution to the field, as determining the optimal DNA input for genome skimming or mitogenome skimming is crucial, I believe that certain aspects of the experimental design could be further refined to enhance the quality of the paper:
2.1 What are the effects of different mitogenome software on the efficiency of mitogenome recovery? Based on our recent analyses, we successfully recovered a full-length mitogenome using mitogeneExtractor, a newly developed software that aligns reference protein sequences rather than DNA sequences. Why did you choose to use BLAST instead of other available software for mitogenome mining? It may be that certain software tools perform better than BLAST for mitogenome skimming.
2.2 As mentioned previously, I recommend including tapestation or gel images of the DNA and DNA libraries used in the study as supplementary material. This would provide readers with an indication of the quality of the input DNA, which is important, as DNA quality can impact sequencing outcomes.
2.3 Why was the number of PCR cycles increased, given that this may result in more PCR duplicates? Could you specify the number of PCR cycles used for each DNA concentration?
2.4. Why include 1pg in the concentrations used when realistically it is not the common case?

Validity of the findings

I believe that the overall experiment and findings are valid. However, I would like to see a more in-depth discussion of genome size and genome complexity of these two zooplankton species in the paper’s discussion section. I am convinced that genome size and genome complexity significantly affect sequencing outcomes aside from just DNA quantity, particularly in the case of Calanus glacialis, which is known for its large repetitive regions. More evidence or explanation would help clarify the extent to which these factors influence sequencing success in this species.

Additional comments

4.1. What are the genome sizes of both species, and what are the implications of a highly repetitive, large genome on the amount of starting DNA input required? Please include in the introduction also aside from discussion.
4.2. Does the success of recovering the mitogenome depend more on quantity rather than quality of libraries prepared than on the initial input material?
4.3. Please discuss the effect of shotgun sequencing on large genomes. Does the amount of sequence data that can be recovered vary based on the type of sequencing method and platform used?
4.4. While genome skimming is a broad term, the focus here is solely on mitogenomes. Are there implications for other genomic regions, and could they also be analyzed through genome skimming?
Specific comments
• Line 52: "from zooplankton species with small body sizes"
• Line 75: Add "with lowering costs": "The rapid development of HTS technology and lowering costs..."
• Line 97: How was C. glacialis identified? Was it based on morphology? What morphological characteristics were used to distinguish it from C. finmarchicus and C. marshallae, both of which are morphologically similar to C. glacialis?
• Line 101: Why were these specific concentrations chosen?

Reviewer 2 ·

Basic reporting

Line 86: replace "reference mitochondrial sequences" with "mitochondrial reference sequences"
Line 279: replace "decreased" with "decrease"
Figure 3B: The mapping does not fit with the sequence of genes in the top row.
Table 1: Capture should be "Information about sampling events", not "Information of samplings".

Experimental design

A weak point is the absence of technical replicates.

Validity of the findings

The lack of technical replicates should be acknowledged in the discussion.

Additional comments

This study is a minor but nevertheless informative contribution to genetic research on environmental zooplankton samples.

Reviewer 3 ·

Basic reporting

The main research question is not well defined. Although the paper comments on the importance of zooplankton identification, which can be readily done using environmental DNA and metabarcoding methods, it is actually focused on genome skimming of two crustacean species and studying the minimum amount of DNA which can be used for HTS.

Raw data are not made available and figures should be improved. The article would benefit from revision by an English native speaker too.

Literature references need to be expanded, and insufficient field background/context is provided.

See for example: Liu P, Lohman GJS, Cantor E, Langhorst BW, Yigit E, Apone LM, Munafo DB, Stewart FJ, Evans TC Jr, Nichols N, Dimalanta ET, Davis TB, Sumner C. A Fast Solution to NGS Library Prep with Low Nanogram DNA Input. J Biomol Tech. 2013 May;24(Suppl):S44. PMCID: PMC3635320.

Experimental design

Investigation was not performed to a high technical standard and methods lack detail and information to replicate.

Validity of the findings

Key underlying data, such as the Qubit quantification values and raw sequencing data were not provided.

---

## Round 0.2 · accepted · Accept

Dear Dr. Hirai,

One of the reviewers agreed to review the revised version of your paper, but never delivered the report. However, I have made sure that the revisions you have submitted carefully and completely fulfil the reviewers' requirements and that all justifications have been properly submitted. Thank you for taking the reviewers' suggestions into account. I therefore consider the paper to be ready for publication and congratulate you on your work.

I wish you all the best,
Olja Vidjak